# Behavioural analysis of postnatal physical activity in the UK according to the COM-B model: a multi-methods study

Kate Ellis, Sally Pears, Stephen Sutton

Department of Public Health and Primary Care, University of Cambridge, Cambridge, UK

**Correspondence to**
Kate Ellis;
nke22@medschl.cam.ac.uk

## ABSTRACT

**Objective** Develop a behavioural analysis of factors influencing postnatal physical activity (PA) according to the 'capability, opportunity, motivation and behaviour' (COM-B) model of behaviour to inform intervention development using the Behaviour Change Wheel (BCW).

**Design** Cross-sectional, multi-method study using semi-structured interviews and a quantitative questionnaire.

**Setting** Children's centres and mother and baby groups in Hertfordshire and Cambridgeshire, UK.

**Participants** Convenience samples of postnatal women were interviewed (n=16) and completed the questionnaire (n=158).

**Methods** Semi-structured interviews followed a preprepared topic guide exploring the COM-B model components and analysed using framework analysis. The questionnaire, based on the self-evaluation of behaviour questionnaire, was adapted using patient and public involvement and findings from the interviews. Questionnaire participants rated their agreement with 22 predefined statements related to COM-B model components. Mean, SD and 95% CI were calculated and each item categorised according to importance. Demographic data were collected.

**Results** The questionnaire identified that new mothers would be more active if they had more time, felt less tired, had accessible childcare, were part of a group, advised by a healthcare professional, able to develop a habit and had more motivation. Additional themes emerging from qualitative data were engaging in PA groups with other new mothers, limited physical stamina following complicated births, social interaction, enjoyment and parental beliefs as motivation, provision of child-friendly PA facilities and environments and babies' unpredictable routines.

**Conclusion** The behavioural analysis presented in this paper identifies and adds detail on the range of factors influencing the target behaviour. Some are unique to the target population, requiring targeted interventions for postnatal women, whereas some are individualised, suggesting the need for individually tailored interventions. We will use the behavioural analysis presented to design an intervention using the subsequent steps in the BCW.

## INTRODUCTION

In the UK, 42% of women do not meet the UK physical activity (PA) guidelines for health,[1] despite well-documented physical and

### Strengths and limitations of this study

► A strength of the research is the use of two data sources and methods to identify factors influencing postnatal physical activity (PA).

► The study links factors influencing postnatal PA to a pre-existing model of behaviour as part of a wider intervention development process.

► Participants were recruited from children's centres, which could predispose our sample to prefer social interaction and group contacts.

► The generalisability of findings may be limited given that mothers were recruited from a single region in the UK and resulted in a demographically homogenous sample. There was low representation from some demographic groups.

psychological health outcomes.[2] Additional positive outcomes of PA for postnatal women within 12 months of childbirth are reduced postnatal depressive symptoms,[3] reduced postnatal weight retention[4] and positive influence on children's PA levels.[5 6] However, the prevalence of postnatal inactivity is concerning,[7] and longitudinal studies show low PA levels throughout pregnancy and the postnatal period.[8 9] Postnatal women are less active compared with age-matched peers, fathers[10] and parents of older children.[11] Encouragingly, research has described this period as a 'teachable moment', a major life event which provides an opportunity for health behaviour change.[12]

Best practice guidelines for developing interventions recommend using theory[13] and suggest that theory-based interventions are more effective than non-theory based. However, theory is not related to intervention efficacy in postnatal PA interventions, potentially because the chosen theories omit important influences on behaviour,[14] or interventions do not target or successfully change theoretical constructs. The Behaviour Change Wheel (BCW) is a comprehensive method for developing interventions, based on a behavioural model applicable to a range

of health behaviours.[14] It has been used to develop interventions to increase provision of PA advice by healthcare professionals,[15] adolescent girls' PA[16] and long-term hearing aid use.[17] The first stage of the BCW involves understanding the target behaviour and culminates in a behavioural analysis, identifying what factors need to change to enable behaviour in relation to the 'capability, opportunity, motivation and behaviour' (COM-B) model. The COM-B model proposes that individuals' capability, opportunity and motivation interact to influence the target behaviour. Capability refers to individuals' physical and psychological capability to engage in a behaviour, comprised of physical capability, having the physical strength or stamina to perform the behaviour and psychological capability, the knowledge or psychological skills, strength or stamina to engage in behaviour. Opportunity refers to environmental factors that influence behaviour and may be physical or social opportunities. Motivation includes all brain processes that guide behaviour and includes reflective and automatic processes. Reflective motivation include individuals' evaluations and plans to engage in behaviour and automatic motivation refers to emotions, impulses and habits.

Existing literature relating to postnatal PA can inform a behavioural analysis;[18] however, previous research has limited participants to report one[19] or four,[20] barriers/enablers thus restricting our ability to determine the range of factors. To explore a broader range, including environmental factors,[21] one in depth qualitative study adopted a socioecological approach. The study identified the key barriers to PA as fatigue, lack of motivation and confidence, time constraints, access to activities and poor public transport and enablers of partner support.[7] The study provided detail to complete the capability and opportunity components of the COM-B model, yet there are limited findings on the motivation component. The paper identified a lack of motivation among participants that could be enhanced by social support; however, there may be additional factors influencing motivation. This paper can inform the behavioural analysis; however, the small sample size may not have reached data saturation, meaning some influencing factors could be omitted. To date, no research has quantified the relative importance of the broad range of factors identified, which could aid intervention designers to ensure they target the most important factors in interventions.

This multi-methods study aims to (1) explore what factors influence postnatal women's capability, opportunity and motivation for PA and (2) identify their relative importance.

## METHODS
This study used a multi-methods design and received ethical approval from the Psychology Research Ethics Committee, University of Cambridge (PRE.2017.037 and PRE.2017.077). We followed the Standards for Reporting Qualitative Research (see online supplementary file 1).

We recruited convenience samples of participants from children's centres, mother and baby groups and online forums in Cambridgeshire and Hertfordshire, UK, by distributing advertising flyers. Children's centres were chosen as the primary recruitment source as 85% of families use their services within the first year of birth,[22] supplemented by online forums to target mums who were not engaged with group activities. Eligible participants were within 12 months of childbirth, aged 16 years or over, lived with their youngest child, were in good general health and spoke sufficient English (interview only) and were ineligible if they had postnatal depression, gestational diabetes during pregnancy or were currently pregnant. Both methods used the same eligibility criteria and recruitment methods but there was no overlap between participants for the two study methods.

### Semi-structured interviews
Participants responded to study advertisements by expressing an interest. We assessed participants for eligibility and, if eligible, arranged a telephone or face-to-face interview. We sent a reminder email prior to the interview and if participants were not contactable for the interview, attempted to rearrange the interview. Participants provided written consent before the interview and received no compensation for participating. The semi-structured interviews followed a preprepared topic guide (see online supplementary file 2) to explore participants' capability, opportunity and motivation, using prompt questions to elicit additional information. Interviews were recorded and the interviewer made field notes. KE conducted the interview who had undergone training and teaching in qualitative methods. We collected demographic data on participants' age, number of children, age of youngest child, employment status, education level and PA levels, measured using the International PA Questionnaire-Short Form (IPAQ-SF). The IPAQ-SF is a widely used self-report tool with minimal participant burden, measuring PA domains applicable to the target audience. The IPAQ-SF shows acceptable validity and reliability.[23]

KE and SP were involved in the analysis process. KE transcribed the interviews verbatim and checked for errors by listening to the audio recording and reading the transcript simultaneously. Anonymised transcripts were imported into NVivo V.11. Framework analysis was used because we were working with predefined themes. KE and SP familiarised themselves with the data and independently coded the first three transcripts. KE and SP met to discuss their coding and link the codes to the preselected themes of the COM-B model and used the initial framework to code the next five transcripts, and met to discuss emerging themes. This process was repeated with subsequent transcripts until no new codes were identified in three consecutive transcripts indicating data saturation. The final coding framework was used to code all transcripts by KE and verified by SP. The data were charted into a framework matrix and interpreted.

**Table 1** Questionnaire development process

| Description of step | Questionnaire changes |
| --- | --- |
| 1. Tailor questionnaire for current study | Adapted existing statements to relate to PA. Adapted the measurement scale to allow participants to rate factors as 'important' or 'not important'. |
| 2. PPI feedback | Panel feedback suggested modifying the measurement scale to 'agree' and 'disagree' allowing statements to be framed positively. Appearance changes to declutter the questionnaire. Change language tone to be warmer and more empathetic. Comprehensive coverage of all influencing factors. |
| 3. Pilot with target population | Questionnaire instructions and statements were clear. Identified childcare and receipt of healthcare professional advice as additional factors to add to questionnaire. |
| 4. Refine using qualitative interview data | Tiredness identified as a theme in interviews to add to the questionnaire. |

PPI, patient and public involvement.

## Questionnaire

We based the questionnaire on the COM-B Self-Evaluation Questionnaire V1,[24] a questionnaire applicable to a range of health behaviours and populations and recommended for use during the BCW intervention development process. We used a four-stage process to develop the questionnaire (table 1). The final questionnaire presented 22 prespecified statements and asked participants to rate the extent they agreed with each statement (1=strongly disagree; 7=strongly agree) (table 2). We collected demographic and IPAQ-SF data, described above.

We calculated sample size by estimating the precision of the mean using the 95% CI. A sample of 130 provides a mean precise to ±0.35, which was deemed acceptable. We distributed paper questionnaires and a hyperlink to the electronic questionnaire, hosted at www.qualtrics.com. Participants provided informed consent prior to participation and received no compensation for completing the questionnaire. Participants completing a paper questionnaire were screened for eligibility using paper copies of the eligibility screening form and returning to the researcher. Eligible participants were given a paper questionnaire to complete. The online questionnaire used skip logic to direct ineligible participants out of the questionnaire and eligible participants to complete the questionnaire.

Data analysis used IBM SPSS Statistics V.25. IPAQ-SF data were processed and analysed according to recommended guidelines,[25] and demographic data were analysed using descriptive statistics. We calculated mean and SD for each statement response and categorised them into agree (≥4.5) neutral (≥3.5 <4.5) and disagree (<3.5).

## Patient and public involvement

We used patient and public involvement (PPI) to adapt the original questionnaire. Fifteen members of Cambridge University Hospitals PPI panel reviewed the questionnaire and provided feedback, resulting in changes to the tone of language and the appearance of the questionnaire. We acknowledged comments about high participant burden

and assessed this during piloting. Three members of a mother and baby group piloted the questionnaire using a think aloud protocol, where participants were asked to verbalise each thought that crosses their mind when completing the questionnaire.[26] The questionnaire was clear and easy to understand and because of their comments, we added statements relating to childcare and advice from healthcare professionals.

## RESULTS
### Semi-structured interviews

Twenty-three participants responded to the study adverts and were screened for eligibility. Three participants were ineligible (currently pregnant n=1; history of gestational diabetes mellitus n=2), and four were not contactable for their interview. Sixteen participants completed the interviews (telephone=4; face-to-face=12). Table 3 presents participant demographic characteristics. Table 4 presents a behavioural analysis based on the qualitative interview data.

### Capability: psychological

Key sources of information were social media, children's centres, online forums, friends from prenatal groups and word of mouth. However, participants did not feel that they were equipped with sufficient information about local opportunities, for example, mother and baby exercise classes, or about how to resume PA safely following childbirth. This could be due to moving to new areas, not receiving advice from healthcare professionals or a lack of available opportunities. New mothers would benefit from information signposting to suitable PA opportunities and safe activities that aid recovery from health professionals.

### Capability: physical

Of the participants who were active, they engaged in a variety of activities, including walking, cycling, swimming, postnatal exercise classes and home gyms, suggesting they are physically capable of PA. However, many of these

**Table 2** Questionnaire statement responses

| Questionnaire statement* *I would be more active if…* | Mean (SD) | 95%CI | Questionnaire response % | | | | | | | Categorisation† |
|---|---|---|---|---|---|---|---|---|---|---|
| | | | 1 | 2 | 3 | 4 | 5 | 6 | 7 | |
| **Capability** | | | | | | | | | | |
| I had a better understanding of why it was important | 2.34 (1.60) | 2.09 to 2.59 | 39.9 | 28.5 | 10.1 | 10.1 | 5.7 | 1.9 | 3.8 | Disagree |
| I knew what to do | 3.43 (1.94) | 3.13 to 3.73 | 22.2 | 17.7 | 13.3 | 12 | 19.6 | 7 | 8.2 | Disagree |
| I were physically stronger | 3.35 (1.90) | 3.04 to 3.66 | 21.5 | 20.3 | 12.7 | 15.2 | 16.5 | 5.7 | 8.2 | Disagree |
| I learnt strategies, eg, goal setting | 3.40 (1.81) | 3.12 to 3.68 | 19.0 | 17.7 | 18.4 | 13.9 | 17.7 | 7.0 | 6.3 | Disagree |
| I did not give up so easily | 3.82 (2.01) | 3.5 to 4.14 | 17.1 | 16.5 | 10.8 | 13.9 | 20.3 | 8.2 | 13.3 | Neutral |
| I had more stamina physically | 3.85 (1.90) | 3.55 to 4.15 | 14.6 | 16.5 | 12.0 | 14.6 | 20.3 | 13.3 | 8.9 | Neutral |
| I had more stamina mentally | 3.85 (1.84) | 3.56 to 4.14 | 16.5 | 12.0 | 8.9 | 20.3 | 26.6 | 7.0 | 8.9 | Neutral |
| **Opportunity** | | | | | | | | | | |
| I had more time | 6.06 (1.46) | 5.83 to 6.29 | 3.8 | 0.00 | 1.9 | 5.7 | 17.1 | 12.0 | 59.5 | Agree |
| I had more money | 4.17 (2.11) | 3.84 to 4.5 | 14.6 | 13.3 | 13.9 | 9.5 | 16.5 | 12.0 | 20.3 | Neutral |
| I felt less tired | 5.61 (1.65) | 5.35 to 5.87 | 3.8 | 2.5 | 5.7 | 10.1 | 15.2 | 19.6 | 43.0 | Agree |
| I had childcare | 5.52 (1.79) | 5.25 to 5.81 | 5.1 | 3.8 | 6.3 | 8.9 | 15.2 | 15.8 | 44.9 | Agree |
| I had the right kit, for example, clothes, trainers, pram | 3.20 (1.87) | 2.91 to 3.49 | 21.5 | 24.7 | 13.3 | 13.9 | 13.9 | 4.4 | 8.2 | Disagree |
| It were easier to access facilities, for example, leisure centres, gyms, swimming pools | 4.37 (1.99) | 4.06 to 4.68 | 10.1 | 14.6 | 7.0 | 18.4 | 16.5 | 13.9 | 19.6 | Neutral |
| There were suitable spaces to be active, for example, public parks, greenspaces, well lit/safe footpaths | 3.85 (1.94) | 3.55 to 4.15 | 15.8 | 15.2 | 10.8 | 17.1 | 19.6 | 10.1 | 11.4 | Neutral |
| **Motivation** | | | | | | | | | | |
| I were part of a group | 4.66 (1.83) | 4.37 to 4.95 | 10.8 | 3.8 | 10.1 | 11.4 | 28.5 | 18.4 | 17.1 | Agree |
| I were prompted to do so | 4.25 (1.80) | 3.96 to 4.52 | 9.5 | 12.0 | 10.1 | 19.0 | 22.2 | 16.5 | 10.8 | Neutral |
| I had encouragement from those around me | 4.34 (1.81) | 4.06 to 4.62 | 8.2 | 10.1 | 12.7 | 21.5 | 17.7 | 15.2 | 14.6 | Neutral |
| I was advised to do so by a healthcare professional | 4.54 (1.96) | 4.23 to 4.85 | 10.8 | 10.1 | 7.0 | 15.2 | 20.3 | 16.5 | 20.3 | Agree |
| I had more motivation | 4.58 (1.87) | 4.29 to 4.87 | 8.9 | 7.0 | 12.0 | 17.7 | 17.1 | 18.4 | 19.0 | Agree |
| I felt it would do me good | 3.68 (1.85) | 3.39 to 3.97 | 17.7 | 12.7 | 13.3 | 22.2 | 16.5 | 10.1 | 7.6 | Neutral |
| I felt I could develop a habit | 4.65 (1.78) | 4.37 to 4.93 | 8.9 | 5.1 | 10.1 | 13.9 | 29.1 | 15.2 | 17.7 | Agree |
| I had a plan | 4.49 (1.87) | 4.2 to 4.78 | 10.1 | 8.9 | 9.5 | 13.3 | 27.2 | 13.9 | 17.1 | Neutral |

*Participants responded on a scale of 1 (strongly disagree) to 7 (strongly agree).
†Mean response to statement categorised as agree ≥ 4.5, neutral ≥3.5 <4.5, disagree < 3.5.

**Table 3** Demographic characteristics

| Characteristic | Interview N | Interview % | Questionnaire n | Questionnaire % |
|---|---|---|---|---|
| **Age (years)** | | | | |
| 16–24 | 2 | 12.5 | 13 | 8.23 |
| 25–30 | 5 | 31.25 | 34 | 21.52 |
| 31–35 | 5 | 31.25 | 75 | 47.47 |
| 36–40 | 4 | 25 | 30 | 18.99 |
| 41–45 | 0 | 0 | 5 | 3.16 |
| 46+ | 0 | 0 | 1 | 0.63 |
| **Age of youngest child (months)** | | | | |
| 0–3 | 1 | 6.25 | 36 | 22.78 |
| 4–6 | 8 | 50 | 52 | 32.91 |
| 7–9 | 5 | 31.25 | 50 | 31.65 |
| 10–12 | 2 | 12.5 | 20 | 12.66 |
| **No of children** | | | | |
| 1 | 14 | 87.5 | 102 | 64.56 |
| 2 | 2 | 12.5 | 47 | 29.74 |
| 3 | 0 | 0 | 6 | 3.80 |
| 4 | 0 | 0 | 1 | 0.63 |
| 5+ | 0 | 0 | 2 | 1.27 |
| **Highest education** | | | | |
| Some secondary school | 0 | 0 | 2 | 1.27 |
| GCSE* | 0 | 0 | 10 | 6.33 |
| A level/equivalent | 8 | 50 | 23 | 14.56 |
| University/college degree | 8 | 50 | 123 | 77.85 |
| **Employment status** | | | | |
| On maternity leave | 12 | 75 | 122 | 77.21 |
| Part time employment | 2 | 12.5 | 10 | 6.33 |
| Full time employment | 0 | 0 | 12 | 7.59 |
| Unemployed | 2 | 12.5 | 14 | 8.86 |
| **Marital status** | | | | |
| Married | 7 | 43.75 | 111 | 70.25 |
| Cohabiting | 9 | 56.25 | 39 | 24.68 |
| Single | 0 | 0 | 6 | 3.80 |
| Separated | 0 | 0 | 2 | 1.27 |
| **Physical activity levels** | | | | |
| Low | 2 | 12.5 | 31 | 19.6 |
| Moderate | 8 | 50 | 62 | 39.2 |
| High | 3 | 18.75 | 28 | 17.7 |
| Excluded † | 3 | 18.75 | 37 | 23.4 |

*UK qualification taken in UK schools at age 16
†Participants excluded due to missing data as advised in IPAQ data processing rules.

activities are different compared with prepregnancy, for example, engaging at a lower intensity or participating in activities that do not require childcare, for example, walking. Some participants report difficulty engaging in particular types of PA such as walking for transportation due to its impact on fatigue or high-impact activities such as spinning resulting in knee pain. Participants acknowledge these limitations and find alternative ways of being active. Participants who had a caesarean section or complicated birth are a subgroup identified with limited physical capability during the early postnatal period, limiting the distance they can walk and specific movements, for example, controlling movement of the pram downhill. This is especially prominent as participants have not received information and are unaware (psychological capability) of how to re-engage in PA. Some have positive beliefs in their physical capability, understanding that they have a reduced physical capability and set realistic expectations such as '*taking it for a mile and then try and build up*'.

### Opportunity: social

Most participants report that their partners are/would be supportive of them to be active, ranging from verbal encouragement, engaging in PA together or providing practical support (eg, purchasing equipment or providing childcare). Despite their intentions, partner support is sometimes limited by work commitments or the baby's reliance on mothers for feeding. Not all new mothers felt supported by their partners with one mother citing the importance of looking at the whole family because it is '*easier to be active if you are both doing it*', or their partners preferred them to be '*getting on doing everything rather than him having to do it*'.

Participants expressed a preference for engaging in PA with another person or a group because it provides accountability and encouragement during the activities to persevere. Specifically, participants preferred groups of new mothers because they are all '*in the same boat*' and create a non-judgemental environment where they appreciate they are '*not going to be looking as you were pre-pregnancy*'. They can share experiences, advice and support specifically relating to motherhood. One participant who originally expressed anxiety towards engaging in group activities welcomed the opportunity to talk to other new mothers about PA, but favoured an approach that allowed relationships to develop organically.

Some participants said their immediate family would be supportive but this was difficult for some because families lived far away, had other family commitments or defaulted to sedentary activities such as a '*cup of tea and a chat*' when they spent time together.

### Opportunity: physical

Physical opportunity is the longest section reported in table 4, reflecting the greatest number of subthemes that emerged from the interview data. Childcare is the key consideration for physical opportunity. Mothers must be satisfied that care is in place for the baby via traditional childcare or engaging in activities that allow her to care for the baby.

**Table 4** Qualitative behavioural analysis

| Definition | Summary | Participant quote |
|---|---|---|
| *Capability: individual's physical and psychological capacity to engage in the behaviour* | | |
| *Psychological: capacity to engage in necessary thought processes* | Need knowledge of local activity opportunities suitable for postnatal women. | 'I wouldn't have found out about Bounce without the other girls…so making sure they're advertised through baby groups or children's centres,' <br><br> P02, Moderately active, 1 child, age 7–9 months |
| | Need to know about easy and manageable activities that are safe and aid recovery from birth. | 'with the recovery and getting back to exercise now, it's all like I want to get there, or I want to get to the first week really, but what's all the steps leading up to that?' <br><br> P05, Invalid IPAQ data, 2 children, age 0–3 months |
| *Physical: capacity to engage in the necessary physical processes* | C-section or complicated births limit some specific movements in the early postnatal period, for example, lifting, pushing a pram downhill. | 'roads are a bit of an issue at the moment pushing the pram because the big movement that I can't do having had the C-section is pushing down on the pram to lift the wheels up to get up a kerb.' <br><br> P05, Invalid IPAQ data, 2 children, age 0–3 months |
| | Reduced stamina following childbirth and need for exercises that aid recovery. | 'It was actually far more tough than I realised, C-Section and getting back on your feet and going for walks. It took me ages actually.' <br><br> P07, Highly active, 1 child, age 4–6 months |
| *Opportunity: all factors lying outside the individual that make performance of the behaviour possible or prompt it* | | |
| *Social: cultural milieu that dictates the way we think about things* | Family and partner provide encouragement to be active. | 'he wants me to go out and he wants me to sort of, spend that time with her, with my daughter and sort of out with other friends just to make sure that I'm not at home on my own.' <br><br> P02, Moderately active, 1 child, age 7–9 months |
| | Postnatal women need opportunity to be part of a group, to create accountability and provide encouragement. | 'If there was a group of people I'd be quite happy to meet up with them, because again it's a social thing. But if I was going on my own I'd do it probably once or twice and then think 'oh I can't be bothered now.' <br><br> P06, Low active, 1 child, aged 7–9 months |
| | Want to be active with other new mothers who understand their current situations. | 'you're all looking a bit flabby and horrible and you don't care cause you're all in it together you know. If I was going to go and join some aerobics class I think I'd feel quite unfit by comparison but cause it's a postnatal class everyone's in the same boat.' <br><br> P04, Moderately active, 1 child, aged 4–6 months |

Continued

**Table 4** Continued

| Definition | Summary | Participant quote |
|---|---|---|
| *Physical: physical opportunity provided by the environment* | Need to ensure that the baby is cared for either through the provision of childcare or PA that involves the baby. | *'then suddenly thought, 'I can't go cause I've got the baby, I just can't do it' and I thought 'noooo'.'*<br>P07, Highly active, 1 child, age 4–6 months |
| | Partners are the main source of childcare, mainly in the evening. | *'I can't get childcare for her unless my partners at home, so as soon as he comes in I would go out to the gym if there was something that I enjoyed,'*<br>P06, Low active, 1 child, aged 7–9 months |
| | Need activity options where they can take the baby, instructor/teacher creates a culture where they feel comfortable to tend to baby's needs. | *'it's really important to be able to bring the baby, because not being able to do that immediately excludes a lot of people'*<br>P07, Highly active, 1 child, age 4–6 months<br>*'like a physical trainer, but maybe a bit more lenient and understanding about mother caring and all that and yeah, I don't know. Someone that knows and like understands it,'*<br>P10, Moderately active, 1 child, aged 4–6 months |
| | Classes should not clash with other activities, should be local and affordable and/or offer flexible payment options. | *'So times conflicting, so when you first have the baby, obviously, you like, you try and do all the activities you can and I always found that all the activities always ended up on the same day, everything would be at the same time on the same day…you pay for sessions, most things you pay for like a block of classes and then the baby is sick for like, a couple of weeks or has really bad diarrhoea and you think 'I can't take him' and you end up missing stuff and everything's really expensive for stuff you don't do.'*<br>P07, Highly active, 1 child, age 4–6 months |
| | Not comfortable leaving babies in a separate crèche. | *'They always seem to be sort of somewhere else, which I wouldn't feel comfortable with, just leaving him in the care of somebody else, I don't know I just wouldn't feel comfortable with that'*<br>P09, Moderately active, 1 child age 4–6 months |
| | Child-friendly activities above are not available, sell out immediately or are too expensive. | *'the only one I found that had spaces, that's the other thing—a few of them are cheaper, but they sell out immediately, were more than £14 a lesson, and I just couldn't justify that kind of price.'*<br>P07, Highly active, 1 child, age 4–6 months |
| | Need accessible and pleasant walking routes—good pavements or footpaths, safe and greenspace. | *'for me to going for walks it's the fact I have nice places to walk, safe places to walk and well lit places to walk.'*<br>P07, Highly active, 1 child, age 4–6 months |
| | Bad weather—rain, dark, cold, puts new mothers off doing activities outdoors, especially when the baby is also exposed to the weather. | *'I didn't go out for a walk the other day because it was raining, and like the effort of having to go upstairs and find the rain covers and knowing how wet it would be and everything else I didn't bother, that was a big impact.'*<br>P05, Invalid IPAQ data, 2 children, age 0–3 months |
| | Breastfeeding means the baby needs to be with mothers at all times, especially during early stages when their feeding routines are unpredictable. | *'I'd feel guilty in case he didn't want to take the bottle and he wanted the breast and then he was crying.'*<br>P03, Moderately active, 1 child, aged 4–6 months |
| | Feel too tired to take part in PA. | *'the last couple of nights he's been up every forty five minutes. Just to get through the day is a challenge so, I just don't have the energy. I don't have the energy until his sleeping improves I just don't have the energy to do extra.'*<br>P03, Moderately active, 1 child, aged 4–6 months |
| | When doing PA, the baby can disrupt the activity, for example, climbing all over them, or crying and being discontent. | *'I wouldn't be able to do anything at home. She's just on the go all the time. She tries to climb my legs, um, so anything like that where I've got to move my legs it wouldn't be possible, because she's up them.'*<br>P06, Low active, 1 child, aged 7–9 months |

*Motivation: all brain processes that energise and direct behaviour*

Continued

**Table 4** Continued

| Definition | Summary | Participant quote |
|---|---|---|
| *Automatic: emotions and impulses arising from associative learning and/or innate dispositions* | Social interaction with other adults is a motivation for being active. | 'having normal conversations with people. Even like walking to the post office and back and saying hi to the person behind the till was important in those early weeks.' P04, Moderately active, 1 child, aged 4–6 months |
| | New mothers need to get out of the house and get some fresh air. | 'being able to get outside, a lot of the time I'm you know in the fresh air, and just enjoying the outside.' P08, Moderately active, 1 child, 4–6 months |
| | Want to take part in PA that is fun and enjoyable. | 'it's about having fun isn't it, as well as exercising. If I'm not having fun, I'm not going to carry on doing it.' P06, Low active, 1 child, aged 7–9 months |
| *Reflective: evaluations and plans* | New mothers understand physical and mental health benefits of PA. | 'it just makes me feel better and it lifts my mood. Just my general, generally happier. I just feel more motivated to do anything and everything. I feel very lazy without it.' P09, Moderately active, 1 child age 4–6 months |
| | Sense of responsibility to set a good example to children and to be healthy as the children grow up. | 'it helps them to see as they grow up that that's what you've got to do. You know, there's no sitting on computers all day' P01, Invalid IPAQ data, 1 child, 10–12 months |
| | | 'I'm 35, so I'm an older mum, and I want to make sure I'm fit and healthy to keep up with her.' P04, Moderately active, 1 child, aged 4–6 months |
| | Time away from the baby creates a fear that they may miss out on child development or 'mum guilt' for leaving the baby when it may need soothing. | 'he's doing all new things at the minute and he's learning things off me, so I think I need to be around him at the minute.' P11, Moderately active, 1 child, 7–9 months |
| | | 'because it's in the evening now, he's a nightmare in the evening and I think I'd just feel guilty leaving him.' P03, Moderately active, 1 child, 4–6 months |
| | Other priorities, for example, baby/family, housework and sleep compete for mothers' time, money and energy. They need to feel that PA is a priority. | 'It's the family time or your exercise and I'd rather spend time with my family than go to an exercise class for an hour.' P11, Moderately active, 1 child, 7–9 months |
| | | 'I could do it if I didn't do some other stuff but then I just feel that's prob... that's more important. Because if that doesn't get done, then that's going to affect me more than if I don't exercise probably?' P09, Moderately active, 1 child age 4–6 months |

Partners are the most common source of childcare and are a practical solution when work patterns allow daytime childcare. Factors such as being too tired and fear of missing family time interrupt evening childcare opportunities. Some mums are reluctant to use childcare, for example, childminder/crèche because they feel concerned leaving the baby at a young age or the additional cost of childcare.

Activity opportunities that allow mothers to care for their babies (eg, mother and baby exercise classes) create a supportive, baby-led environment where they feel comfortable to tend to the baby's needs, such as feeding, soothing, changing and keeping the baby entertained. These opportunities also provide social interaction with other mums, cited as a key influencing factor in social opportunity. Time, location and cost of activities influence access to these activities. Opportunities must not clash with other local classes, for example, those provided by children's centres, need to be local to mums, preferably within walking distance and offer low cost and flexible payment options where mothers do not lose their money if they cannot attend a session (often due to unavoidable reasons such as baby illness). Walking and cycling are activities that allow mothers to care for their babies when the environment is conducive to walking and cycling. Environmental facilitators are good walking surfaces, safe spaces, well-lit spaces, access to greenspace and facilities (eg, coffee shops or baby changing). Bad weather is a barrier for postnatal women because it also exposes the baby to the cold/wet weather.

Even when care is available, the baby can be a barrier to PA. A lack of routine during early postnatal period means unpredictable feeding and sleeping times; consequently planning PA is difficult. Additionally, unexpected events (eg, baby being sick or in an unsettled mood) may disrupt plans. Disrupted sleep causes feelings of tiredness and as the babies grow up, it is difficult to engage in PA while caring for the baby as they are moving about '*climbing up my legs*'. Participants who breastfeed are especially reluctant to leave the baby because they believe they are the only ones who can soothe the baby or they prioritise using the store of expressed milk for other occasions.

## Motivation: automatic
Enjoyment emerged as a key aspect of participants' automatic motivation. Choosing to engage in activities perceived as fun and enjoyable means they are more likely to maintain participation. Seeing the baby enjoy and having a laugh are all elements that contribute to participants' enjoyment of PA. A second automatic motivation is the opportunity to '*get out of the four walls*' and view going for a walk as a chance to get outside and get some fresh air. Building on this, participants desire social interaction, which '*can be as small as saying hi to the person behind the tills in the post office*', having adult conversations or developing friendships with other '*like-minded people*'. This is a way of overcoming social isolation noted by some participants.

## Motivation: reflective
Collectively participants demonstrate a comprehensive understanding of the physical and mental health outcomes of PA. However, individual participants were not aware of the whole range of benefits. Unique to this population is the contribution that PA can make to their role as a parent: first, by creating an active culture within the house and role-modelling healthy habits; second, making them feel refreshed and ready to deal with everything; third, to promote long-term health as their children grow up; and lastly, to improve preconception health for future pregnancies. These can be counterbalanced by negative parenting beliefs, for example, missing family time and child development milestones, 'mum guilt' when spending time away from the baby and guilt that no one else will be able to soothe the baby. One mother was concerned that engaging in PA would make her tired and impact her parenting ability. Possibility of injury resulting from PA was also a concern for some participants.

Despite an overall positive evaluation of PA in isolation, when considered in a wider context, there are competing priorities for limited time, money and energy. Participants say that '*everyone else comes before you*', or '*something's gotta give and certain things need to get bumped off the checklist*'. The value that participants place on each competing priority determines which activities get '*bumped off the checklist*'. Some value being active and will place this above competing behaviours '*because of me pushing my own exercise routine, my household is suffering*', whereas others prioritise the competing behaviours '*if (the other stuff) doesn't get done, then that's going to affect me more than if I don't exercise*'.

## Questionnaire
Two hundred and eighty-eight participants responded to study advertisements. There were 99 incomplete responses and 31 ineligible participants (>12 months since childbirth n=10; currently pregnant n=3; poor general health n=4; history of gestational diabetes n=7; experiencing postnatal depressive symptoms n=6; not living with baby n=1). One hundred and fifty-eight participants completed the questionnaire. Table 3 presents participants' demographic data.

Table 2 presents participant responses to the questionnaire statements. Participants would be more active if they had more time (mean=6.06; 95% CI 5.83 to 6.29), felt less tired (mean=5.61; 95% CI 5.35 to 5.87), had access to childcare (mean=5.52; 95% CI 5.25 to 5.81), were part of a group (mean=4.66; 95% CI 4.37 to 4.95) and were able to develop a habit (mean=4.65; 95% CI 4.37 to 4.93). Participants would not be more active if they understood why PA is important (mean=2.34; 95% CI 2.09 to 2.59), had the right kit (mean=3.20; 95% CI 2.91 to 3.49), felt physically stronger (mean=3.34; 95% CI 3.04 to 3.66), learnt strategies (mean=3.40; 95% CI 3.12 to 3.68) and knew what to do (mean=3.43; 95% CI 3.13 to 3.73).

## DISCUSSION

This study found that postnatal women would be more active if they had more time, were less tired, had access to childcare, were part of a group, received advice from a healthcare professional, had more motivation and could develop a habit. At face value, these are broadly similar to the general population,[27 28] with the exception of childcare provision. The additional detail gained from the qualitative data demonstrates aspects that are unique to the target population. For example, tiredness is exacerbated by the baby's disrupted sleep patterns, preference for group activities with other new mothers, motivation is enhanced by positive outcome expectations relating to the baby and combating social isolation and difficulty developing habits because of the babies' disrupted sleeping and feeding routines. This evidence strengthens the case for targeted interventions for the postnatal population. Population level interventions (eg, group-based exercise, healthcare professional advice) should include strategies to address the factors identified as important on a population level in the questionnaire and use the added detail from the qualitative data to ensure content is applicable to new mothers. Knowing what to do, money and access to facilities and suitable spaces were identified as factors in the interview but not the questionnaire. One explanation is that some factors are individualised, showing a need for individually tailored interventions, as suggested elsewhere.[7]

In this study, data from two sources are complementary; the qualitative data provide detailed explanations of the factors influencing postnatal PA and the questionnaire quantified these factors to determine their order of importance. For example, the questionnaire identified time as the key barrier, but the qualitative data enabled us to uncover two possible explanations. Participants perceived PA as time consuming or it reflects the priority placed on PA. Active participants prioritised time for PA at a cost to competing behaviours, whereas inactive women prioritised other activities (eg, sleeping and housework). This suggests that the perceived value of PA may determine behaviour, or perhaps that active participants have made a habit of prioritising PA. Participants' questionnaire responses indicated that they wanted to be part of a group, and the interview responses enabled us to understand that groups offer accountability and social interaction. Specifically, participating with other mothers is preferred because of shared experiences, a sense of group cohesion and non-judgemental attitudes. Mother and baby groups are especially attractive because in addition to social interaction, they alleviate the need for childcare and are suitable for breastfeeding mothers. The questionnaire statements did not specifically assess group activity with other new mothers as it was not identified during questionnaire development. Current research on group exercise suggests that perceiving other group members as similar increases attraction and level of involvement with the group,[29] which could explain our participants' preference for mother and baby groups.

To our knowledge, this is the first study to investigate postnatal women's motivations for PA. The questionnaire responses indicate that participants would be more active if they had more motivation, and the qualitative data enabled us to identify enjoyment, social interaction and 'getting out' as contributing to automatic motivation, which may explain our participants' attraction to group-based activities. Interestingly, existing postnatal PA interventions have utilised flexible delivery methods (eg, SMS,[30 31] telephone,[32] websites,[33] apps[34]), following formative research expressing a preference for minimal face-to-face contact and no interest in or inability to join exercise groups.[30 35] However, our sample was recruited from children's centres that provide non-PA related mother and baby groups, potentially predisposing our sample towards social interaction and group activity. Participants' reflective motivation included not only physical and mental health but also baby and parenting related beliefs, reflecting a shift in women's focus after birth to consider the baby in everything they do.[7] Efforts to increase PA in this population could include strategies to enhance positive evaluations and reduce negative evaluations relating to the baby.

The results of this paper categorise findings according to the COM-B model, an existing model of behaviour, linked to an overarching framework to guide intervention development. The present study used the self-evaluation questionnaire[24] which enables participants to consider a wide spectrum of factors relating to the COM-B model. Previous research has limited participants to a finite number of influencing factors,[19 20] which may have limited our understanding. While these studies also found time, tiredness and childcare to be important, our study has identified additional reasons for participants' motivations for PA. The present study follows the BCW methods for identifying influencing factors, using a recommended questionnaire tailored for the target population and using multiple data sources. The authors state that consistency between data sources provides confidence in the results,[24] but we believe the inconsistency in our results is an added insight that has allowed us to understand the influencing factors at a population level and at an individual level. This study revealed interplay between the COM-B model components when investigated in the qualitative interviews. For example, some participants cited limited physical capability because their expectation for postnatal PA was unrealistic and they engaged in too much too soon. This could be a psychological capability deficit as they are unaware of appropriate activities. Healthcare professionals could address this deficit by educating or providing PA advice during one of the multiple contacts during the postnatal period. Despite such interplay, the current model categorises the factors influencing postnatal PA according to the COM-B model components to complete stage 1 of the BCW. Moving this work forward, we will follow the subsequent two steps of the BCW intervention development process to choose intervention options and intervention content.

The BCW pathway links each step, encouraging users to consider and appraise the range of appropriate intervention functions and content.[24] A key element of the resulting intervention is delivering tailored interventions, due to the individualised nature of influencing factors, enabling interventions to focus on the key factors influencing individual participants.

The strengths of this research include the use of two data sources, as discussed above. Additionally, the qualitative research informed the questionnaire development, which may have reduced inconsistencies between the two data sources. The qualitative findings resulted in including childcare, tiredness and healthcare professional advice in the questionnaire, three of the most important factors identified by participants. The questionnaire sample size exceeded the calculated sample size, thus providing precise estimates of group means that we can confidently use to judge the relative importance of the influencing factors. The recruitment of a relatively small sample from a specific context is likely to reduce the information power of the data.[36] Recruiting both active and inactive women allowed us to understand the barriers that prevented inactive participants and the enablers that helped active participants. With regard to study limitations, as mentioned above, we recruited from children's centres and our participants may be those who prefer to engage in social/group activities. The generalisability of this study is limited due to an under-representation of less-educated, single and multiparous women recruited from a single region in the UK. The study does not account for regional variations in the provision and accessibility of services and has limited data relating to the cultural differences in attitudes to PA. Due to the online advertisement of the questionnaires, we were unable to determine response rate for the questionnaire or to identify any demographic differences between responders and non-responders. Research on influencing factors would benefit from including under-represented groups, and future research should use the behavioural analysis presented to develop evidence-based interventions using the BCW.

## CONCLUSION

In this paper, we have presented a comprehensive behavioural analysis, which provides a detailed account of the range of factors that influence postnatal women's capability, opportunity and motivation to be physically active as a base for developing theory-based interventions using the BCW. Influencing factors can apply at a population level (eg, time, tiredness, childcare), while others are applicable at an individual level (eg, knowing what to do, money and access to suitable facilities and spaces), therefore future behavioural interventions should design interventions targeting the appropriate factors when designing population and individual level interventions. Future research should investigate methods to identify what factors influence individuals' PA levels and how to use these to deliver of tailored interventions.

**Twitter** @BSG_Cambridge

**Contributors** KE and SS were responsible for planning and design of the study. KE was responsible for data collection. KE and SP were responsible for data analysis. KE wrote the first draft. All authors read and approved the manuscript.

**Funding** This study is funded by the National Institute for Health Research (NIHR) School for Primary Care Research. The views expressed are those of the authors and not necessarily those of the NIHR, the NHS or the Department of Health and Social Care.

**Competing interests** None declared.

**Patient consent for publication** Not required.

**Ethics approval** The study received ethical approval from the Psychology Research Ethics Committee, University of Cambridge (PRE.2017.037 and PRE.2017.077)

**Provenance and peer review** Not commissioned; externally peer reviewed.

**Data sharing statement** No additional data are available.

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
