## [Reviewer comments · BMJ Open]

ARTICLE DETAILS

TITLE (PROVISIONAL)	A behavioural analysis of postnatal physical activity in the UK according to the COM-B model: A multi-methods study
AUTHORS	Ellis, Kate; Pears, Sally; Sutton, Stephen

VERSION 1 - REVIEW

REVIEWER	Emily F Gregory Children's Hospital of Philadelphia
REVIEW RETURNED	01-Feb-2019

GENERAL COMMENTS	This manuscript addresses an important question, namely barriers & facilitators of physical activity in the postnatal period. I have recommended minor revisions for this manuscript. However, as described below, this manuscript addresses a question that is relatively narrow in scope, which the editors may think warrants rejection. Strengths of this manuscript include: 1. Laying out the importance of the postnatal period for long-term health for women, and potentially their children.2. Providing a clear theory-based rationale for the need for the current study.3. Discussing the complementarity of the methods in a concrete and compelling manner.4. Use of a validated scale and tools in multiple aspects of the research.5. Very clearly written throughout. The weakest aspect of this manuscript, if it could be considered a weakness, is that it addresses a relatively narrow question. This is due to two factors in the research: 1. The authors identify a small gap in prior work – namely that barriers and facilitators to physical activity in this period have been described previously but without allowing an open-ended response scheme that may generate previously overlooked factors.2. The authors conducted this research with a relatively narrow sample. They recruited from community networks already serving new mothers in a single region of the UK, leading to a demographically homogeneous sample. Several specific comments: Page 5, line 52: Please clarify that the existing literature specifically addresses postnatal women.
---

	Methods: Please explain how a determination was made to stop interviews after completing 16. Page 6, line 36 – 45: May want to say something descriptive about the expected population from the sites chosen for recruitment & why they were selected. Page 7, line 4: Please clarify whether participants received any compensation for participation in interviews. Page 7, line 16: Please clarify why IPAQ-SF data was collected. It was clearly used to characterize the physical activity level of participants. However it seems that some interesting analyses could have compared participants across levels of physical activity in both interviews & questionnaires. Why weren't such analyses attempted or described? Given the level of detail the manuscript uses in relating findings to the COM-B model, it may be helpful to better describe the constructs in the model early in the manuscript, perhaps when the development of the questionnaire is described (page 7, from line 43). Otherwise, the first time the reader encounters the construct categories is when results are presented and several of these categories are never explicitly defined. Page 9: Please clarify whether participants completed consent and / or received compensation for the questionnaire. Page 9, line 54: Please describe the "the think aloud protocol" for readers who are not familiar with this method.
--	--

REVIEWER	Dr Rhiannon Phillips Cardiff University, United Kingdom
REVIEW RETURNED	15-Feb-2019

GENERAL COMMENTS	This manuscript describes an important public health issue; novel approaches to reduce inactivity for mothers of young children have the significant potential to improve health outcomes for this population. A strength of the study is the use of a systematic approach, the application of the COM-B model as part of the Behaviour Change Wheel framework, to identify potential determinants of inactivity, taking into account broader environmental as well as individual and interpersonal factors. The importance of capability and opportunity highlighted in the results was of note. The findings of this study suggest that behavior change strategies focusing on provision of information or development of skills alone may not be the most fruitful approach with this population. Many of the barriers to physical activity identified are practical in nature (e.g. accessibility, cost, provision of childcare), whereas facilitators appeared to be more focused on motivational and social factors. While these findings are novel and interesting, this study provides only part of the jigsaw, and intervention(s) developed based on these findings alone are likely to have the greatest relevance to the specific context in which these data were collected. The study certainly indicates the need for further research, but the conclusions
---

should not be over-stated. Specific issues that would need to be addressed to make this manuscript suitable for publication are:

1. The authors should refer to the COREQ-32 checklist (http://cdn.elsevier.com/promis_misc/ISSM_COREQ_Checklist.pdf) and ensure that all aspects of the qualitative study methodology have been fully reported.
2. Acronyms in abstract (COM-B, PPI, PA) should be written out in full at first use.
3. P.5 line 59 and p. 6 line 3 –parentheses can be removed as this is important information.
4. The exclusion criteria mean that key groups were omitted from the study who are at particular risk of low activity benefits and may particularly benefit from intervention (e.g. those with long-term health conditions and depressive symptoms). Justification for the selection of these exclusion criteria should be provided.
5. Further information should be provided on the IPAQ-SF questionnaire and why this particular measure was selected.
6. There was a systematic approach to developing the questionnaire, which included user-involvement. However, the data presented make it difficult to judge how meaningful some of the findings are as there is little information about the measurement properties or validity of the questionnaire. It would be useful, for example, to know more about the range and distribution of responses. It is also unclear why the responses have been categories as disagree, neutral, and agree rather than using the 7-point scale (with the additional sensitivity this offers) during analysis.
7. References should be provided to support the statement 'At face value, these are broadly similar to the general population, with the exception of childcare provision', p. 21, line 15-17.
8. P. 21, lines 42-47, 'One potential method for tailoring interventions is using the questionnaire developed for this study to identify the factors influencing individual behaviour'. This statement should be removed or amended. Though the questionnaire may be useful in this context, it has not been tested in this way, and its validity has not been adequately tested to warrant recommending the measure could be used in this way at this stage.
9. The qualitative data in this study is presented very concisely and in a logical fashion. However, some elements of the findings lack depth, and this is touched upon in the discussion (p. 22), indicating that some of the detail and context has been lost in some places. Women's individual lived experiences are important here – e.g. why do some women seem to find time but others not? Is this because they have less help with the housework for example, given that inactive women cite housework as a barrier. Although I appreciate the need to be concise, some of the most prominent barriers/facilitators could be further explored to get into the finer details of the drivers of activity/inactivity.
10. The qualitative data focus on a relatively small group of women recruited in a specific context. Could the authors comment on the likely 'information power' (see Malterud et al. 2015) of this sample? This needs to be taken into consideration when interpreting the findings.
11. P.22- there could be an important difference between mother and baby groups and other activity groups as mothers can take babies with them to the former, which can be an important factor for those who are breastfeeding or are not able to get childcare in particular.
12. Care is needed in reporting the conclusions/implications of this research. The generalizability of the findings may be limited, not

	only due to the omission of under-represented groups already noted, but also as a result of regional variation in the provision and accessibility of services, and cultural differences in attitudes towards participating in physical activity.
--	--

VERSION 1 – AUTHOR RESPONSE

Please clarify that the existing literature specifically addresses postnatal women. I have added text to clarify this point.

Please explain how a determination was made to stop interviews after completing 16. We stopped interviewing after 16 participants because we coded three successive transcripts where we did not identify any new codes indicating data saturation therefore we made the decision to stop recruitment. I have clarified this point in the methods.

May want to say something descriptive about the expected population from the sites chosen for recruitment & why they were selected. We selected children's centres as the primary recruitment source because a longitudinal evaluation of their use found the service is used by 85% of families within the first year of birth, fulfilling our key eligibility criteria of recruiting participants within one year of childbirth. This was supplemented by online forums as a high proportion of mums are internet users. This information has been included in the manuscript.

Please clarify whether participants received any compensation for participation in interviews. I have clarified in the manuscript that participants received no compensation for participation in the interviews.

Please clarify why IPAQ-SF data was collected. It was clearly used to characterize the physical activity level of participants. However it seems that some interesting analyses could have compared participants across levels of physical activity in both interviews & questionnaires. Why weren't such analyses attempted or described? We used IPAQ-SF to collect PA measures as it is a widely used questionnaire which includes domestic and travel domains which are applicable to the target population. The IPAQ-SF is a short questionnaire with minimal participant burden, yet no difference in reliability and validity statistics compared to the long form. I have included this information in the manuscript. In the manuscript, we have presented the IPAQ-SF as descriptive statistics of the sample (Table 3). It would be possible to conduct and report further statistical analysis in the manuscript to identify whether the factors influencing PA differ across participants with differing PA levels. This would increase the length of the manuscript, however we are happy to include this according to the preferences of the editor.

Given the level of detail in the manuscript uses in relating findings to the COM-B model, it may be helpful to better describe the constructs in the model early in the manuscript, perhaps when the development of the questionnaire is described. Otherwise the first time the reader encounters the construct categories is when the results are presented and several of these categories are never explicitly defined. Thank you for noting this omission, which we also feel is important. I have included an explanation of the COM-B constructs in paragraph two of the introduction, when the COM-B model is first introduced.

Please clarify whether participants completed consent and / or received compensation for the questionnaire. Participants provided consent to complete the questionnaire and received no compensation for completion. This has been included in the manuscript.

Please describe the "think aloud protocol" for readers who are not familiar with this method. I have described and provided a reference of the think aloud protocol.

The authors should refer to COREQ-32 Checklist and ensure all aspects of the qualitative study methodology have been reported. I apologise I had not clearly referred to the reporting checklist that was included in the original submission. I had submitted a SRQR checklist for qualitative research as recommended in the BMJ Open author guidelines. I have revised the manuscript to refer to the SRQR checklist to be published as a supplementary online material. Should the editors wish us to complete a COREQ checklist as opposed to the SRQR, we would be happy to amend this.

Acronyms in abstract (COM-B, PPI, PA) should be written out in full at first use. Thank you for pointing out this mistake – I have corrected this in the manuscript.

Parentheses can be removed as this is important information. I have removed the parentheses.

The exclusion criteria mean that key groups were omitted from the study who are at particular risk of low activity benefits and may particularly benefit from intervention (e.g. those with long-term health conditions and depressive symptoms). Justification for the selection of these exclusion criteria should be provided. We understand the concerns that we have omitted an important population. This study is part of a wider programme of work with an original aim to develop a brief intervention for increasing postnatal PA. We decided to omit these groups as it is likely that they have needs that cannot be met by a brief intervention. While not the focus of the current study, it is certainly something I am very keen on researching in the future!

Further information should be provided on the IPAQ-SF questionnaire and why this particular measure was selected. Please see above comment to reviewer 1.

There was a systematic approach to developing the questionnaire, which included user-involvement. However, the data presented make it difficult to judge how meaningful some of the findings are as there is little information about the measurement properties or validity of the questionnaire. It would be useful, for example, to know more about the range and distribution of responses. The original questionnaire is recommended by Michie and colleagues as a tool to identify what influences behaviour as part of the Behaviour Change Wheel process of intervention development. The original questionnaire is applicable to a range of health behaviours and populations and requires adaptation to use with specific behaviours and populations. Thus there is no information available on the measurement properties of the questionnaire. From the data in this study, I have included additional information in Table 2 to show the response percentage for each point on the scale for all questionnaire items to display the range and distribution of the responses. It is also unclear why the responses have been categories as disagree, neutral, and agree rather than using the 7-point scale (with the additional sensitivity this offers) during analysis. Table 2 presents the mean response on the 7-point scale for all items offering the sensitivity referred to. The categories disagree, neutral and agree were created to help interpret the findings for readers to identify the factors that lie towards the 'agree' end of the 7-point scale and those towards the 'disagree' end of the scale. We feel this adds to the analysis of the 7-point scale. However, should the editor prefer the categories were removed, we would be willing to change this.

References should be provided to support the statement "At face value, these are broadly similar to the general population, with the exception of childcare provision." Thank you for noting the omission of a reference. I have corrected this and included a reference for the statement.

"One potential method for tailoring interventions using the questionnaire developed for this study to identify factors influencing individual behaviour." This statement should be removed or amended. Though the questionnaire may be useful in this context, it has not been tested in this way, and its validity has not been adequately tested to warrant recommending the measure could be used in this way at this stage. We understand these concerns and have removed the statement from the manuscript.

The qualitative data in this study is presented very concisely and in a logical fashion. However, some elements of the findings lack depth, and this is touched upon in the discussion (p. 22), indicating that some of the detail and context has been lost in some places. Women's individual lived experiences are important here – e.g. why do some women seem to find time but others not? Is this because they have less help with the housework for example, given that inactive women cite housework as a barrier. Although I appreciate the need to be concise, some of the most prominent barriers/facilitators could be further explored to get into the finer details of the drivers of activity/inactivity. Thank you for this comment. Where possible we have included additional depth and detail to the qualitative results on page 20-24.

The qualitative data focus on a relatively small group of women recruited in a specific context. Could the authors comment on the likely information power (see Malterud et al, 2015) of this sample? This needs to be taken into consideration when interpreting the findings. The small sample and its specificity to a small region in the UK is likely to reduce the likely information power of the study and this has been included in the manuscript alongside a reference to the paper for readers who are unfamiliar with the concept.

There could be an important difference between mother and baby groups and other activity groups as mothers can take babies with them to the former, which can be an important factor for those who are breastfeeding or are not able to get childcare in particular. This is true and a finding from the research that we wanted to include in the paper. Thank you for pointing out that this was not expressed clearly. We have added a statement which explains this point.

Care is needed in reporting the conclusions/implications of this research. The generalizability of findings may be limited, not only due to the omission of under-represented groups already noted, but also as a result of regional variation in the provision and accessibility of services, and cultural differences in attitudes towards participating in physical activity. Thank you for this useful comment. I have amended the limitations to reflect the limited generalisability to avoid overstating the results.

VERSION 2 – REVIEW

REVIEWER	Emily F Gregory Children's Hospital of Philadelphia, United States
REVIEW RETURNED	09-Apr-2019

GENERAL COMMENTS	The authors have responded to reviewer concerns in this revision and have clearly stated that data draws from a population that may be homogenous in terms of geographic location, preference for socializing and group activities, and other demographics. Without providing page by page examples, I do think the authors could revise this manuscript to make the it more concise without sacrificing depth of analysis. I do think this would contribute to the ultimate readability of the manuscript.
---